# Indigenous Land-Based Approaches to Well-Being: The *Niska* (Goose) Harvesting Program in Subarctic Ontario, Canada

**DOI:** 10.3390/ijerph20043686

**Published:** 2023-02-19

**Authors:** Fatima Ahmed, Eric N. Liberda, Andrew Solomon, Roger Davey, Bernard Sutherland, Leonard J. S. Tsuji

**Affiliations:** 1Department of Physical and Environmental Sciences, University of Toronto, Toronto, ON M1C 1A4, Canada; 2School of Occupation and Public Health, Faculty of Community Services, Toronto Metropolitan University, Toronto, ON M5B 2K3, Canada; 3Fort Albany First Nation, Fort Albany, ON P0L 1H0, Canada; 4Peetabeck Academy, Mundo Peetabeck Education Authority, Fort Albany, ON P0L 1H0, Canada

**Keywords:** Indigenous, First Nations, well-being, cortisol, stress, goose harvesting, food security, photovoice, two-eyed seeing, wellness, subarctic Canada

## Abstract

Historically, goose harvesting provided a source of culturally significant, safe, and nutritious food for the *Omushkego* Cree of subarctic Ontario, Canada. Disruptions stemming from colonization and climate change have led to a decrease in harvesting, resulting in higher rates of food insecurity. The aim of the *Niska* program was to reconnect Elders and youth to revitalize goose harvesting activities and associated Indigenous knowledge within the community. The program and evaluation were built using a two-eyed seeing (*Etuaptmumk*) and community-based participatory research approach. Salivary cortisol, a biomedical measure of stress, was collected before (*n* = 13) and after (*n* = 13) participation in the spring harvest. Likewise, cortisol samples were collected before (*n* = 12) and after (*n* = 12) the summer harvest. Photovoice and semi-directed interviews were employed after the spring (*n =* 13) and summer (*n =* 12) harvests to identify key elements of well-being from an Indigenous perspective. The changes observed in cortisol levels for the spring (*p* = 0.782) and summer (*p* = 0.395) harvests were not statistically significant. However, there was a noteworthy increase in the subjective well-being observed through the qualitative measures (semi-directed interviews and photovoice), highlighting the importance of using multiple perspectives when assessing well-being, especially in Indigenous peoples. Future programs should incorporate multiple perspectives when addressing complex environmental and health issues, such as food security and environmental conservation, especially in Indigenous homelands worldwide.

## 1. Introduction

The increasing threat of climate change has had many detrimental impacts on public health globally [1]. One example of this is the impact on food security, which is the ability to access safe and nutritious food that meets the dietary needs of individuals for an active and healthy lifestyle [2,3]. This growing threat has disproportionately impacted Indigenous communities, especially those living in remote areas [4,5]. Food systems in these communities are either comprised of traditional food, which has been obtained through harvesting or sharing, or market foods, which are bought in stores [4]. Traditional harvesting practices which are deeply rooted in familial and social systems of Indigenous communities continue to be negatively impacted by colonization and colonial assimilative policies (e.g., residential schools) [2,6,7,8,9]. This disruption has contributed to the growing issue of food insecurity amongst Indigenous peoples in Canada [3,4,10].

Understanding the disproportionate impacts of food insecurity and the benefits of food programs based on Indigenous knowledge is particularly important for remote Indigenous communities. Fort Albany First Nation (FN) is a remote *Omushkego* Cree community which is only accessible year-round by plane, by boat during ice-free seasons, and by ice roads after a freeze-up. Its location makes the transport of goods including food very expensive, as the nearest entry point for food distribution is in Timmins, Ontario, which is 769 km south of Fort Albany FN [5]. These factors contribute to the growing rate of food insecurity within the community. A study by Skinner, Hanning and Tsuji [5] estimated that 70% of households were food insecure, with a 76% prevalence in households with children. Some of the main factors contributing to this are the increased cost of food, increased dependence on market foods, and the prohibitive costs (i.e., equipment) of participating in traditional land-based activities such as, hunting, trapping, and fishing [4,5,11,12]. Although the importance of traditional foods is widely recognized, a large portion of the dietary intake is from store-bought foods, especially for young people in the community [5,11,13]. The readily available prepacked, store-bought food is known to be higher in refined sugars and saturated fats [14], and could potentially lead to higher rates of obesity, diabetes, and cardiovascular disease [3,15,16] as adults. Indeed, higher rates of obesity for the youth on-reserve—43% being overweight or obese, versus 26% of the youth in the general population [17]—has been reported. Although the quality of the diet is an important factor contributing to the overweightness and obesity, the adoption of other aspects of western lifestyles (e.g., increasing sedentary behaviors, decreasing amounts of physical activity) has also been implicated [17].

Subsistence activities are an integral part of the *Omushkego* Cree culture. An example of this is the harvesting of the Canada goose (*Branta canadensis*) in the spring, which is seen as a celebration of life and survival through the winter [18]. Goose harvesting promotes social and community cohesiveness and has long provided a source of nutrient-rich, relatively inexpensive food [3]. A study by Gates, Hanning, Gates and Tsuji [17] found that moderate consumption (2–6 times per week) of snow goose *(Chen caerulescens)* was associated with higher intakes of protein, vitamin B_12_, iron, and zinc. Traditionally, many factors have impacted the access to and availability of traditional foods, with the overall participation in harvesting activities declining in many communities [17]. Factors such as climate change, including unpredictable weather, have been affecting the pattern of migratory birds within the James Bay region [3,18,19,20,21,22]. Populations of lesser snow geese *(C. caerulescens caerulescens)* have increased significantly since the late 1960s due to anthropogenic factors, such as climate change and the decline in the number of hunters [3,10,23]. The foraging activities of these lesser snow geese have been observed to have caused significant damage to arctic and subarctic ecosystems by increasing the soil salinity and altering moisture conditions, leading to desertification [23,24]. The western area of James bay is particularly vulnerable to the impact of the lesser snow goose because of the intensive feeding in thawed wetlands that have minimal to no above-ground vegetation growth [23]. Another species harvested in this area is the molt-migrant giant Canada goose (*Branta canadensis maxima*). Giant Canada geese were once extinct in Ontario, and were reintroduced in 1965 [18]. During molt-migration, these geese move from more southerly areas to more northerly areas to molt, that is, shed old feathers for new flight feathers [25]. These giant Canada geese not only cause agricultural loss in the south but also compete with other migratory birds for food in the north [18]. Their feces, which contain *Escherichia coli*, have been shown to contaminate drinking water reservoirs and can be a serious risk to human health [26].

It is important to note that the levels of organochlorines and toxic metals have been shown to be relatively low in both the giant Canada goose [3,27,28,29,30] and the lesser snow goose [3,27,28,29], making them relatively safe for human consumption. Increased harvesting of these species would allow for more opportunities for the habitat to recover [18]. Historically, harvesting of the snow goose for the *Omushkego* Cree was done in the fall; however, springtime harvesting allows for the removal of reproductive adults from the population [3,18], thereby further impacting the population density and aiding the conservation efforts with respect to subarctic and arctic ecosystems [3,10,17,23,24,27,28].

Recently, an overall decrease in the goose harvesting and time spent on the land has been observed in Fort Albany FN and the James Bay region [3,16,18,23], with many factors contributing. One of the main factors for this decrease relates to the cultural changes associated with colonization, including the disruption of the intergenerational transfer of Indigenous knowledge [3,5,7,10,18,31,32]. This knowledge is crucial to maintaining practices, such as goose harvesting which offers many benefits, in the community. Although harvesting practices have become easier in some respects because of technological advances such as snowmobiles, more efficient firearms and ammunition, and modern refrigeration and freezer units [18], these advances have also led to a disruption in the familial and community aspect of harvesting practices [3,18].

Climate change is another major factor which has been impacting the participation in goose harvesting activities. It is well known that the effects of climate change are being felt disproportionately at higher latitudes, leaving northern Indigenous communities more susceptible to its impacts [20,33,34,35,36]. The average temperatures in the arctic have increased by nearly twice the global average over the last century, with regional observations indicating a thinner ice cover, increasing variability in the weather patterns, changes in the wind and precipitation, along with warmer seasons [20,22,34,37]. In Fort Albany FN, community members have observed warmer springs and summers, along with milder winters which are also of a shorter duration [22]. This has also affected inland ice conditions, which impact the safety when traveling on the ice for subsistence activities [3,20,38]. Another environmental change being observed in the James Bay region is glacial isostatic adjustment, a natural phenomenon related to the uplift of land after the release of the weight of the ice sheets, after the glaciers melted [39]. In the Fort Albany FN region, this has caused sea levels to recede even with global warming [37]. These environmental changes have all been cited as factors which have required continuous adaptions by community members to ongoing changes in the environment in this region [22,28].

Indigenous populations, especially those living in rural or remote communities, interact closely with their environment and therefore have developed specialized knowledge and practices that not only offer valuable insight, but they are also adaptive to shifts and changes in the environment [40]. This knowledge is being more readily acknowledged as fundamental to addressing the impacts of climate change on ecosystem management [1,10,22,34,41] and also human health and well-being [1,7,22,42,43,44]. Within a Canadian context, the Community Well-Being Index was developed to measure well-being across Indigenous and non-Indigenous communities, using data on income, education, housing, and labor force activity [45]. Communities are given scores ranging from a low of zero to a high of 100, which then aim to reinforce the importance of community-specific approaches when developing policies and programs. Notably, this index does not include culturally relevant variables, such as the relation to the land, which has been cited as an important indicator of well-being for Indigenous peoples [13,33,46,47]. The absence of these factors indicates a need for research into further understanding health and well-being in the context of social and cultural factors, and beyond the traditional biomedical approach [47]. Methods such as photovoice, which engages and empowers individuals through reflection and discussion of their intra- and interpersonal experiences, helps to give a better understanding of these social and cultural factors. Another promising method in examining health and well-being is the quantification of salivary cortisol, a biomedical measure of stress. Chronic stress over a long period can negatively impact an individual’s health and well-being, resulting in a variety of negative health outcomes. Many Indigenous communities experience stress related to socioeconomic factors, disease, intergenerational trauma, the environment, and other health and social inequities [48,49,50,51]. Climate change [52] and colonization [31] have individually been identified as factors impacting the allostatic load, which can alter cortisol concentrations due to the cumulative burdens of chronic stress. The unequal exposure to stressful social and environmental factors impacts how the brain processes stimuli and has been shown to underlie health and well-being disparities, with stress being of importance [13,49,51]. There is a need for more studies that employ both the Indigenous and western perspectives to address the complex factors influencing the health and well-being of Indigenous populations.

Within Fort Albany FN, Elders and other community members have expressed the need for more programs following the seasonal harvesting practices of the *Omushkego* Cree to ensure the cultural continuity and increase the social cohesion [13]. One such program was the sharing-the-harvest initiative [3], in which the benefits of geese harvesting on food security, environmental conservation, and social cohesion were identified. The impacts of geese harvesting on well-being from both a biomedical and Indigenous perspective have yet to be examined. For these reasons, the *Niska* (goose) harvesting program was developed to address barriers to participation and to further our understanding of the impacts of geese harvesting on individuals and the community. Using complementary biomedical (i.e., salivary cortisol) and Indigenous perspectives (i.e., photovoice and semi-directed interviews) allowed for participants to guide the program whilst sharing their perspectives and experience. The overall aims of the goose program were to not only address the issues of food security and environmental conservation, but to also contribute to the health and well-being of participants and the continent-wide initiative to harvest over-abundant goose species.

## 2. Methods

The methods for this study were similar to the *Amisk* (beaver) harvesting program [13], which sought to address the issues of community flooding, an overabundance of beavers, and the intergenerational disruption of Indigenous knowledge within Fort Albany FN of subarctic Canada. Similarly, this *Niska* (goose) harvesting program was developed to address community concerns related to the decreased participation in harvesting activities. The following sections briefly summarize the methods and additional information may be found in Ahmed, Liberda, Solomon, Davey, Sutherland and Tsuji [13].

### 2.1. Study Area

The James Bay region of northern Ontario is part of the Mushkegowuk Territory, which is home to the *Omushkego* Cree who inhabit several First Nations communities that are only accessible by air and a seasonal winter road. This road, referred to as the James Bay winter road, is located around the western coast of James Bay. The community of Fort Albany First Nation FN (Figure 1) is located within this region, with approximately 1400 people registered [53] and 900 people living on-reserve (i.e., living in the community).

The community is located in one of the largest wetlands and muskeg (bog or swamp) areas of the world [20]. It is also located near many major water bodies, such as James Bay, Moose River, Attawapiskat River, and the Albany River, which is one of the largest rivers in the province of Ontario [32]. These water bodies have served as a means for transportation and have a deep-rooted cultural and historical significance for the community. The subarctic climate of the region also means that there are longer winters and shorter summers. The location and climate of this region have provided an ideal habitat for many game species important to the Cree [18,22].

### 2.2. Study Design and Participants

This program was developed, in its entirety, with the Fort Albany First Nation leadership (A.S.) and community-based coordinators (R.D., B.S.). A quasi-experimental study design was incorporated because the goose program was conceived and run by Fort Albany FN, including participant recruitment. The Fort Albany FN Band Council representatives selected the Elders and other on-the-land experts who led the initiatives, as well as participating youth. Elders and on-the-land experts planned the activities, such as when, where, and how the harvesting would take place, and what information would be shared with the non-Indigenous research team members.

Participants were all ≥18 years of age and were required to possess a valid Possession and Acquisition License (PAL) to allow for the legal purchase and possession of firearms and ammunition. Youth recruitment was done by word of mouth through the Band Council representatives, with Elders and on-the-land experts being recruited based on their experience with the traditional harvesting activity. Within Fort Albany FN, the population structure is opposite to non-Indigenous populations, with only 4.3% of the population above the age of 65, 58.6% between 15 and 64 years, and 37.9% below the age of 14 [54]. This, along with the specific criteria for this experience and a valid PAL, were factors which impacted our participant numbers. Only non-lead ammunition was used for this project, as lead ammunition has been identified as a major source of wildlife and environmental contamination [3], and a human health risk [55].

This program utilized the two-eyed seeing *(Etuaptmumk)* and community-based participatory (CBPR) approaches [56]. Two-eyed seeing is an integrative approach which stresses the importance of viewing issues through both mainstream western and Indigenous worldviews for the greater good of everyone [56,57,58]. The second approach, CBPR, allows for practical problem solving when addressing social issues through planning, action, and reflection, and recognizing communities as a unit of identity to build on their strengths and resources [59,60,61]. The use of these complementary approaches provided a better understanding of the health and well-being, whilst allowing for the inclusion of the diverse perspectives and experiences of community members. Additionally, OCAP^®^ (Ownership, Control, Access, and Possession) principals established how the data and information were collected, protected, used, and shared.

Informed consent was obtained in a manner which was inclusive, transparent, and in collaboration with Fort Albany FN. The process was conducted in English, with the option for translators if Cree was a preferred language. Participation in all aspects of the research were entirely voluntary with the option to withdraw at any time and to opt-out of any part of the research (i.e., not taking photos or videos). Participants selected a random number (1 to 37), which remained consistent throughout their participation in other seasonally run programs [13] and allowed for the differentiating of their data (i.e., samples, interviews, and photovoice). Data were kept anonymous, with participants only being identifiable by the principal investigator and community coordinator. Primary ethics approval was obtained through the University of Toronto ethics committee.

#### Goose Harvesting (Spring and Summer)

The duration of the spring 2018 spring and summer harvests ranged from 2 days to 1 week. Participants were all male and ranged from ages 21 to 84 years. Salivary cortisol samples were taken the morning prior to commencing the activity and the morning after completion of the activity. Elders and on-the-land experts led youth along the Albany River and its tributaries, where youth were taught methods of goose hunting; Elders and on-the-land experts determined the types of firearms which would be used, the location, the timing of the harvest, and provided instructions on safety measures. Motorized cargo canoes, snowmobiles, and all-terrain vehicles (ATVs) were used for transportation. The species of goose obtained during the harvests was not tracked for this pilot program. All geese obtained through this program were shared by participants with the community.

### 2.3. Data Collection and Analysis

#### 2.3.1. Salivary Cortisol

Salivary cortisol was collected using Salivette^®^ cotton swab tubes (Sarstedt, Numbrecht, Germany) in the mornings upon waking, prior to consuming any food, drink, or smoking, and prior to and after participation in each activity. Samples were stored in a community freezer by the community coordinator and then at the University of Toronto laboratory where they were stored at −20 °C.

The cortisol samples were then analyzed by In-Common Laboratories (Toronto, ON, Canada), which is a medical testing laboratory licensed by the Ontario Ministry of Health and Long-Term Care, and accredited by the Institute of Quality Management in Healthcare and holds the 15189 PlusTM certificate. ElectroChemiLuminescence Immunoassay (ECLIA) analysis using Cobas e 411 Analyzer (Roche Diagnostics GmbH, Mannheim, Germany) was completed, with the detection limits for concentrations being Inter-assay coefficients of the variation at high and low concentrations of 12.53% and 8.98%, respectively, with an intra-assay coefficient of variation not being available.

Statistical analysis was completed using the Statistical Package for Social Sciences (SPSS) 13.0 (SPSS, Inc., Chicago, IL, USA). Normality testing was done using the Shapiro–Wilk test and then log transformed, with a paired *t*-test (two-tailed) to examine if differences between pre- and post- concentrations were statistically significant. Similar to Ahmed, Liberda, Solomon, Davey, Sutherland and Tsuji [13], which followed Jung*,* et al. [62], samples below the detection limit of 1.5 nmol/L were input as such, with a *p* value < 0.05 being considered significant.

#### 2.3.2. Photovoice and Semi-Directed Interviews

Photovoice was used to empower individuals by providing an opportunity to present their perspectives and experiences with regards to well-being. A GoPro Hero5 (GoPro, Inc., San Mateo, CA, USA) was provided to each participant and they were instructed to take photos of anything they associated with their well-being or wellness, with narrations following. Participants who did not take photos were still invited to participate in the semi-directed interview, which consisted of questions (Appendix A) that were formulated with the FN Advisory Committee.

For both measures, participants could bring an English/Cree translator of their choice if they wished, with English translations used for the interviews done in Cree. Interviews were conducted in a neutral setting and recorded on data recorders. All data were stored securely, physically, and electronically, at the University of Toronto on encrypted computers, which are only accessible to the authors. Interviews and narrations were manually transcribed verbatim using the NVivo 11 (QSR International Pty, Melbourne, Australia) software analyzer. Using a deductive approach, we followed the theoretical framework from our previous study [13]. The thematic analysis included multiple stages of reviewing the transcriptions to identify phrases, sentences, or paragraphs of importance. Coding these sections allowed us to determine relationships between themes under the main categories from the previous study (knowledge, identity, healing, and land) and further allowed for the development of subthemes by activity.

## 3. Results

### 3.1. Salivary Cortisol

The results of the within-group comparisons of salivary cortisol are displayed in Table 1 and Figure A1, Figure A2, Figure A3 and Figure A4 (Appendix A). Of the total 22 participants, only 1 declined to participate in sampling for the spring and summer projects. A total of 54 samples were collected for the spring and summer projects and 2 samples were omitted as they were not quantifiable. The pair-wise exclusion of samples also excluded any pairs where the post-activity samples were not quantifiable, resulting in a final total of 50 samples.

#### 3.1.1. Goose Spring Harvest

The mean difference between the pre- and post-participation cortisol levels was −0.55 ± 0.71 nmol/L, which was not statistically significant (*t* (12) = −0.28, *p* = 0.782, two-tailed) (Figure A2).

#### 3.1.2. Goose Summer Harvest

The mean difference between the pre- and post-participation cortisol levels was 0.78 ± 0.30 nmol/L, which was not statistically significant (*t* (11) = 0.89, *p* = 0.395, two-tailed) (Figure A4).

### 3.2. Semi-Directed Interviews and Photovoice

A total of 25 semi-directed interviews and narrations for both the spring (*n* = 12) and summer (*n* = 13) were completed, with a 28% overlap in participation of the harvests. Three participants were unable to participate in the interviews for various reasons. The interviews and narrations ranged from 5 min to 1 h. The themes presented in the subsequent sections are supported by verbatim quotes which captured the stories, emotions, and opinions of the participants. The descriptive along with the analytical themes are presented in Table 2. It should be emphasized that the themes generated were not all mutually exclusive. Figures presented in the subsequent section were censored and any personal identifiable information (name, age, etc.) from the quotes was omitted to maintain anonymity.

Quotes for the summer program were in concordance with the spring program results and therefore, for brevity, were placed in Appendix A (Table A1). The narratives presented in the subsequent sections helped to increase our understandings of the individual and community health and well-being, along with the revitalization of traditional goose harvesting practices.

#### 3.2.1. Goose Harvesting Spring

This program took place in the spring of 2018 and provided an opportunity for participants to take part in the harvest of Canada geese and lesser snow geese. Many participants discussed the importance of the program to them from both an individual and community perspective,

“*I would like to see this continue going… [the] program, you got to do to help the youth out… It’s always going to be good thing if you help somebody not to get hungry… That’s the most important thing… I lived through it, and it got me in trouble… I just moved on, I kept going and I just push myself*”.(Participant 20, Elder)

##### Knowledge

The importance of sharing knowledge amongst familial, community, and other social circles was a reoccurring theme. Sharing this knowledge during the harvesting practices with each other, regardless of experience level, allowed participants to engage in the continuity of the Cree culture. Participants expressed positive emotions, including an eagerness to partake in the goose harvest to share and obtain knowledge:

“*There’s (name omitted) (Figure 2), he’s a residential school survivor. You learn a lot of things from that guy*”.(Participant 16, Youth)

“*Where I lack knowledge, I can learn more from an older person*”.(Participant 33, Expert)

As experienced participants and Elders led the direction of the programs, it allowed for a richer experience for youth, who were exposed to a diverse set of knowledge and skills.

Disseminating the knowledge that they have obtained, no matter what their level of experience was, was seen as an opportunity to better themselves and their community:

“*Run traditional ceremonies, the whole point is to become a teacher and to maybe one day teach about healing… Maybe one day someone is struggling with drugs or alcohol*” .(Participant 16, Youth)

“*Living off the land it’s a good thing for everybody cause everything is free out there you don’t have to buy nothing. It’s just up to you to do it… You have to encourage the youth to do that… but they need us at the same time… We need to have them follow us… I don’t know what’s going to happen… Once the government stops feeding us, then who’s going to feed you now? That’s the danger, cause right now we don’t have gardens… Nothing to sustain us for food… if something really happens, we’ll starve*” .(Participant 20, Elder)

“*You need to teach the young people when you take them out on the land. Sometimes you ask them, it’s so boring they say, but if you start teaching them how to listen… Being there in May, it’s the best time to take the kids out, and all these things are starting to come out… When you come back in June and these things are growing, this will be a medicine… You have to be able to recognize it when it grows… what it looks like when it’s full grown… also recognize it when after it’s done its job… You should be able to say that was the medicine that that I took home and used*” .(Participant 3, Elder)

An important factor impacting the knowledge dissemination within the community is the declining use of the Cree language, because of colonization and the residential school system. Many participants found it difficult to converse with Elders and on-the-land experts who spoke high-level Cree, as compared to conversational Cree or no-Cree at all:

“*It’s not like the old days… you can’t really take out the young people anymore because of language barrier… The Elders are passing away. They’re the only ones that speak Cree mostly like 24/7… That’s the first part of the culture cause without language you can’t talk to your Elders, right?*” .(Participant 36, Elder, English Translation from Cree)

“*I like this thing that’s going on, taking out young people… My dad talks to them and sometimes they understand, but they can’t speak… I’m like a middle person to translate, so it works out perfect*” .(Participant 13, Expert)

He later went on to discuss learning opportunities stating, *“The only way you can learn your language, keep your language alive, it’s to visit your Elders. Talk with them… speak our language at home”* (Participant 13, Expert). Although this barrier was present, it did not hinder participation in the program, as those who required it were able to partner with translators or learn by watching:

“*Well-being is like practicing your knowledge… then passing it down to someone else. It’s the way you teach somebody… He’s just looking at you… he’ll be able to figure it out… doing things by example eh? If I shoot a caribou and I start gutting it out and if it has a young caribou, then the first thing I do is… remove the young caribou… I take it out and I put it to the side and then I make a peace offering… I don’t say it, I just do that… Later on, I’ll talk about it right… at our tent, I’ll sit around, and I’ll say that I made a peace offering because I took a caribou and released the other spirit… Ever since I can remember our Elders did it… it’s always been done since time memorial, we don’t really know how it began but we still do it today*” .(Participant 3, Elder)

Many of the images and statements described the importance of knowledge dissemination as an essential part of individual and cultural well-being for participants, the community, and to sustain future generations.

##### Identity

Taking part in the harvest allowed participants to reflect on their Cree identity and how goose harvesting and spending time out on the land impacted it. As stated by one on-the-land expert, *"That’s where I grew up all my life, that’s where I was born… and that’s why I like it”* (Participant 12, Expert). Others described how hunting and obtaining traditional food were vital aspects of their life:

“*He did that all his life, trapping and hunting and it was good for his health, and he wouldn’t get sick cause he was always out in the wild, fresh air… He didn’t sit around or anything like that he was pretty active*” .(Participant 36, Elder, English Translation from Cree)

“*I got it from my old man… Taught me traditional way how to hunt… I was right out there. That’s how I learned… that’s how I mostly survived on, was wild meat, sometimes well store*” .(Participant 17, Expert)

“*I need hunting to have a good life, to go with our traditional ways… It’s important cause that’s how we lived our life*” .(Participant 23, Youth)

For many, taking part in goose harvesting not only allowed for the development of experiences and skills which have shaped their identity, but also allowed them to establish important connections with youth and other community members (Figure 3).

The connections established during the goose harvesting provided a positive experience for youth, furthering their cultural and individual well-being.

#### 3.2.2. Cultural Continuity

A subtheme of identity was the notion of cultural continuity, which is the ability to preserve and carry forward traditions. For some youth, the opportunity to take part in goose harvesting was a first for them. Learning these essential skills was a way to maintain their culture and ultimately improve their own well-being, especially for those struggling with issues of food insecurity or substance abuse:

“*For me… it would be to not get too much involved with… drugs, alcohol, bootlegging… it’s not well-being… Well-being would be go hunting and do your culture stuff… Teach kids how it’s done… Very important in their life if they don’t have that confidence then they’ll seem out of place or they’ll go in the wrong direction*” .(Participant 20, Elder)

“*Nobody does this anymore… That’s what they did all the time to survive. Go out on the river somewhere and live… That’s what I do for a living eh? Hunt*” .(Participant 12, Expert)

The social aspect of these activities was cited as an important factor which should be leveraged to increase participation in activities, 

“*When you have opportunities like this, take it… It’s a good experience, you’ll have fun out there with yourself and maybe with other people, like friends or especially with, family… Being out there you could have good memories*” .(Participant 16, Youth)

“*There is always opportunities… This program is an opportunity… sometimes you create your own… You wanna help young kids so you teach them something, a life skill for example… Learn it and teach other people and they’re going to continue*” .(Participant 3, Elder)

It is clear from the discussions that an important aspect of maintaining the *Omushkego* Cree culture was the idea of reciprocity and taking care of one another. As stated by one participant, *“To sustain my dad, with his adventure and his activities out there. I still do it; I’ve been hunting with him since I was 10 years old… keep on going where he’s use to take me as a child… Sometimes I take my nephews”* (Participant 13, Expert). 

Elders and experts shared the importance of sharing knowledge with the youth to help them to overcome some of the challenges when participating, *“They don’t have no camps, eh? They can’t go in the bush… They weren’t born that way”* (Participant 12, Expert). Technological advances were acknowledged as having many benefits, but were also seen as a barrier to well-being and cultural continuity:

“*These young generations. They are not really how I grew up. I was hooked on… hunting, fishing… since my childhood… People are starting to lose interest… We’re in a new century… Everything’s all easy for them. Everything’s all technology now*” .(Participant 17, Expert)

“*That guy there, he had this there that he can connect online out there… You know sometimes you got to leave your things behind… You’re somewhere else when you do that… you’re not aware of all the surroundings… You’re too busy spending time on them [devices] and you’re not recognizing your medicines*” .(Participant 3, Elder)

“*When we stopped my dad told me to tell him to put that thing away… his phone… He told me to tell him look around*” .(Participant 36, Elder, English Translation from Cree)

It was evident that goose harvesting was important for the cultural continuity along with well-being, as it increased the social cohesion and allowed for more opportunities for the transfer of Cree knowledge.

##### Healing

Participants frequently spoke of different types of healing (mental, physical, emotional, spiritual) which took place while participating in goose harvesting activities. The social aspects of the activity, along with the increased knowledge and experience, gave individuals a great avenue to improve their well-being. Participants spoke about their emotional and mental health when on the land:

“*I’m brand new again, I’m revived… it clears your mind”* .(Participant 23, Youth)

“*It’s just so calm out there it’s just like an invisible medicine… It’s very healthy for the mind*” .(Participant 33, Expert)

“*I feel free when I’m out there, but here I kinda feel trapped*” .(Participant 6, Youth)

As goose harvesting was also a very physically demanding activity (Figure 4), many participants spoke about how it was an opportunity to improve their physical well-being:

“*He’s sitting here the whole time, he’ll get sick… He’d go in the bush and in two days he’d get better… To be healthy, it’s just to go out there*” .(Participant 36, Elder, English Translation from Cree)

“*Healthwise, to stay active… walking and working, keeping your mind busy*” .(Participant 13, Expert)

Although the land is seen as an opportunity for healing and improving well-being, there were many challenges and barriers which were identified, some of which stemmed from colonization,

“*What happened in the past it’s still going on today. What happened in residential schools… I don’t know what they were thinking. Trying to get the Indian out of us… We should have more events, more of this kind of thing [the program]*” .(Participant 20, Elder)

Another challenge within the community was substance abuse, with many speaking of its impacts at an individual and community level:

“*They’re (Elders) slowly passing away and not everybody knows the land and the rivers… For the young people you can’t really teach them anymore because of the drugs and alcohol… It clouds their mind*” .(Participant 36, Elder, English Translation from Cree)

“*Drugs and alcohol when they came, they started messing up our traditional ways*” .(Participant 23, Youth)

“*We can’t really go out together anymore because of… break and enters… too many drugs and alcohol in the community and one has to look after the fort*” .(Participant 13, Expert)

Although these issues were prevalent; many stated that participating in the harvest and other traditional activities were a means of beginning their healing journey:

“*I don’t plan on going back there to my old life because I struggled a lot with drugs… doing speed made me feel depressed… I actually had three overdoses… Right now I’m on my healing journey… I haven’t done any drugs for over two months now… I didn’t even smoke cigarettes too, so I’ve been cold turkey for two and a half months*” (Participant 16, Youth)

Overall, participating in harvesting was seen as a means of improving their well-being in all areas of health such as mental, physical, spiritual, and emotional. It was seen as a means of overcoming barriers and challenges which have negatively impacted them and the community. The social aspects of harvesting, along with the individual development of skills to sustain themselves, were all factors which contributed to the health and well-being of participants.

##### Land

The land was a continual theme present in all the discussions, which highlighted the strong connection each individual had with the land. When asked what was important to their well-being, many frequently referenced the land or nature:

“*I like it out there, there’s peace and quiet. You’re out there with nature. It’s great being out there… I prefer the land but everything’s kind of here you know*” .(Participant 17, Expert)

“*That would be one of the best opportunities you could get. Getting wild meat or fish… I went on the boat on the weekend, I came home, and I slept all night… There’s a lot of fresh air out there, and there’s less stress… Your mind is clear… There’s a calmness there… I notice my blood pressure, is kinda high when I’m around here*“ .(Participant 33, Expert)

“*We can get out of society for a bit and feel better… if you go out for a night or so, you already feel better… I enjoyed it cause I love hunting geese*” .(Participant 23, Youth)

The positive phrases and emotions used reiterate the importance of going on the land as an integral part of the *Omushkego* Cree culture.

#### 3.2.3. Food and Other Resources

A subtheme that was evident when discussing the land was the accessibility of food and other resources. Goose harvesting was a practice which provided a source of traditional food, which was seen as healthier and more accessible:

“*Just look at my stomach man, that’s how I look at it… eating properly, not having an empty stomach… It’s kind of important in my culture. I gotta look at my family… make sure they don’t starve and keep doing that for the whole year*” .(Participant 6, Youth)

“*It’s good to eat wild meat… Not like from the store, lots of chemicals in there. They can hurt you… Give you diabetes… In the bush it’s free, good water, fresh air*” .(Participant 12, Expert)

“*You can tell by… the way people want food, and a lot of people at times here are mostly hungry… I can see people getting hungry all before the end of the month maybe before welfare comes. So, to have this kind of thing happening there (the program)… not just geese but caribou hunting, it would be good, fishing and moose hunting*” .(Participant 20, Elder)

“*There’s a lot of lack of outdoors… like parents taking their kids out camping… to learn their culture, or just to go experience… our traditional ways of life… People are relying on the stores… Nowadays the young generations are more reliant on buying stuff… I want them to know their culture*” .(Participant 17, Expert)

“*You’re going to harvest animals for your health… If you assist somebody, or if you harvest food and share it with other people (Figure 5) … you’re raising their well-being. They’re happy that you’ve given them something that you harvested… Most Elders will understand that it’s a lot of work when you hunt eh… So, for you to get them and you to share, they show a lot of respect, and they’ll say thank you very much*” .(Participant 3, Elder)

It was apparent that the access to traditional food was a key component of their health and well-being. As Fort Albany FN is a remote community, food must be flown into the community, therefore making it more expensive. As discussed by one on-the-land expert, *“You suffer… the cost of everything, cause there’s no road”* (Participant 20, Elder). The cost of living, including the cost of food and other resources, was one of the many barriers discussed by participants to their well-being:

“*There’s always gonna be barriers… because of the way things are up here… There’s a lot of people who can’t get canoes… Cause there’s a lot of people that don’t work… not much work up here… Most of the people here are on social assistance*” .(Participant 33, Expert)

“*They don’t really have boats… or people to take them out there*” .(Participant 14, Youth)

“*They don’t have what they need… a lot of them they wanna go, but then they don’t have anything to use to go. It would be good if it was, the way we were… if they did that with the younger ones, and set them up*” .(Participant 20, Elder)

Reducing the barriers for participation was one of the goals of the program. Therefore, it was essential that participants were provided with the resources that they needed. For many, going back to traditional ways of harvesting was a potential solution to address these barriers, *“Way back… my grandfather travelled up the whole James Bay by foot… When he was a young man and back then there were no skidoos. They used to run to Moosonee, they use to work on the ships out in the bay… that was like way back about 1940”* (Participant 33, Expert).

One on-the-land expert discussed an initiative that he and other community members wanted to take to supply more resources to the youth:

“*I was wishing, to make a camp… up the river along the traplines there, or up the trail… We know them by heart… I almost thought of having a bingo one time just to buy skidoos… and toboggans for the kids. I was gonna do that to have fundraising… I’m starting a little bit now with (name omitted). He’s looking after taking the kids out*” .(Participant 20, Elder)

Overall, the benefits of participating in the goose harvesting outweighed the challenges and barriers, which when addressed, increased participation from the youth. Community members showed a willingness to help address these barriers by working together, especially with others who may not have had the same opportunities. They acknowledged that these opportunities would allow some to return to the traditional *Omushkego* Cree ways, would address the present-day issues of food insecurity, and would provide a setting for the transfer of intergenerational knowledge, healing, and a way to reconnect with their culture and protect future generations.

## 4. Discussion

The *Niska* (goose) program aimed to revitalize the community harvesting activities to reconnect youth with Elders and experts, the land, and cultural traditions, while also addressing food security and ecosystem conservation. Utilizing complementary constructs, such as CBPR and the two-eyed seeing approach, allowed us to further our understanding of health and well-being from a biomedical and First Nations’ perspective.

### 4.1. Health and Well-Being

One of the aims of this program was to examine the impacts of goose harvesting activities on the well-being of participants from a remote subarctic FN community. Even though our intervention showed positive effects with respect to the subjective well-being of participants, no statistically significant differences were observed in short-term salivary cortisol concentrations (i.e., stress response) after taking part in the goose harvesting activities. Participants described the importance of traditional activities to the community and how harvesting activities and their overall well-being has been impacted because of colonization and the residential school system. In the literature, the forced assimilation to a different (i.e., western) culture, known as acculturation, has impacted Indigenous communities over the past century [63]. These assimilative efforts in Canada, such as legal assimilation, the residential school system, the “sixties scoop”, and environmental assimilation, have shown to have negative impacts [8,9,31,64]. Indigenous peoples living in northern communities have been particularly vulnerable to this shift away from a primarily subsistence way of life [4]. This shift was observed by participants in our study, especially amongst the youth, impacting their overall health and well-being primarily through food accessibility and mental wellness. When speaking of these impacts, participants frequently referred to them as sources of stress.

Several studies have noted that Indigenous populations have, and currently experience, more stressful life events compared to non-Indigenous populations [1,51,64,65,66]. This stress response is an important biological mediator between psychological and social factors [48,51] and can be potentially analyzed through the analysis of cortisol [67]. Stress triggers the activation of the hypothalamic–pituitary–adrenal axis, resulting in the production of glucocorticoids in the adrenal cortex which bind to glucocorticoid receptors [49]. In situations of acute stress, this leads to what is known as a “fight or flight” response, which leads to physiological changes such as increased heart rate [49]. Measuring salivary cortisol at-home for Indigenous populations has been reported to result in valuable data and high participant compliance, as it offers lower burdens and disruptions to their daily lives versus in a lab setting [68]. Nonetheless, in our study, cortisol results were inconclusive. Perhaps other factors need to be considered when measuring cortisol [49,66,69,70,71,72,73,74], especially in Indigenous populations.

Many of the health inequities associated with Indigenous peoples have been as a result of complex social and environmental factors which converge on chronic psychosocial stress [49]. There has been increasing evidence demonstrating a link between the chronic activation of the biological stress response, also referred to as the allostatic load, and metabolic syndrome, cardiovascular diseases, substance abuse, and mental wellness [48,49,51,65,68,69,71,75,76]. Although our study only measured short-term changes with pre- and post-cortisol concentrations, evidence has shown that one of the conditions leading to allostatic load is repeated “hits” or activations from multiple stressors which leads to a lack of adaptation over time [48]. The experiences of Indigenous populations in Canada related to colonization, the residential school system, and systemic racism have led to intergenerational trauma; these types of stressors can lead to the loss of well-being as described by participants in the present study.

Many participants described that their health and/or wellness dramatically improved when they were on the land; participants also reported experiencing more positive emotions (i.e., relaxed, happy, free, calm) while on the land compared to being in the community. An important stressor which was discussed was trauma, specifically intergenerational trauma because of the residential school experiences. The effects of residential schools, such as the decline in participation in traditional activities, were still being observed in the community. Through discussions, we were able to identify some of the impacts of intergenerational trauma, especially with the youth, when discussing issues of substance abuse and a lack of familial structure contributing to the decreased participation in traditional activities. For this reason, studies with Indigenous and other marginalized communities must consider the accumulation of stress across generations due to deeply rooted historic trauma stemming from colonization, which have led to present-day stressors (i.e., poverty, discrimination, unemployment) [68].

The findings from our study and others [74] highlight the importance of incorporating multiple perspectives when addressing health and well-being. This type of approach has been referred to as two-eyed seeing (*Etuaptmumk*) [58] and is becoming more prevalent in health-based research involving Indigenous populations [57,77,78,79]. Two-eyed seeing refers to seeing from one eye with the strengths of Indigenous knowledge and the other with the strengths of western knowledge but, using both eyes together to address complex issues [56,58,79]. Indigenous knowledge relies on oral traditions which have been passed intergenerationally and includes information about Indigenous identity, culture, history, and the environment. As stated by many of the Elders, the disruption to the intergenerational transfer of knowledge within the community has been impacting many youth. Therefore, on-the-land programs are crucial for the dissemination of Indigenous knowledge. From an Indigenous perspective, the individual, familial, community, and societal responsibilities must be balanced to maintain health and well-being [77]. Within the context of our study, solely focusing on the biomedical measures would not have allowed for the inclusion of the different experiences and the well-being benefits observed by participants.

When discussing well-being, participants shared that taking part in the goose harvesting allowed them to revitalize cultural practices, promote healing, increase their time on the land, and have more opportunities for knowledge dissemination. Other studies involving Indigenous populations in Australia [80,81,82,83], the United States [84,85,86], and Canada [3,5,13,18,22,87,88] have also noted these positive outcomes of culturally relevant programs. The land was not only seen as a source of physical activity and food, but as a setting where healing occurred, with many speaking of feeling less stressed on the land versus in the community. One study by Foulds*,* et al. [89] found that community-based physical activity interventions led to an increase in the overall physical activity and reduction in waist circumference. Participants in our study frequently discussed being more active when on the land, with many stating a noticeable difference in their health when they do not go out on the land (i.e., more sickness, poorer sleep, not eating well). Many related their time spent on the land to a measurement of their health, understandably so with the many positive outcomes they experienced.

Participants also spoke of the negative health outcomes they have experienced as a result of a decline in the harvesting activities. The reduced transmission of knowledge in the subarctic James Bay region is well documented as a concern of Elders and on-the-land experts [13,18,22,37,40,41,44,90]. The loss of Indigenous languages across Canada has been well documented [91]. Although the loss of language was identified as a barrier for many participants, the ability to have a translator of their choice or be partnered with someone who spoke both English and Cree helped to address this barrier. During discussions, Elders and experts also expressed the need for more activities directed towards youth, to provide them with the skills and encouragement to take part in land-based activities. This was also to empower the youth, so they could have the confidence to sustain themselves and their families from the land and teach others what they have learned. One study by Skinner*,* et al. [92] investigated barriers and supports for healthy eating and physical activity in a remote FN community, and identified empowerment as a core issue that must be taken into account when designing health interventions. As stated by Skinner, Hanning and Tsuji [5], the unique characteristics of each community should be taken into account, with the need for local information to direct policies and programs.

The results of the present study were comparable to the predecessor of this program, the *Amisk* (beaver) harvesting program. The *Amisk* program aimed to address community concerns of local flooding by connecting the Elders with youth to revitalize traditional beaver harvesting and introduce other activities, such as the removal of the beaver dams [13]. Participants in the *Amisk* program identified many positive outcomes, for instance, the sharing of knowledge, and the reconnecting of youth with their identity and the land [13]. The cortisol results of the beaver harvesting activities were similar to that of the *Niska* (goose) program, as no statistically significant changes were observed in the salivary cortisol levels pre- and post-activities; however, a statistically significant increase in cortisol levels was observed when breaking the beaver dams. The potential reasons for this increase were the increased physical activity [93] and an immediate “achievement” experienced by participants when the dams were cleared [94]. Overall, discussions with participants identified positive experiences when taking part in both programs, with many explicitly stating that it contributes to individual and community well-being. Engaging the participants through photovoice promoted a critical in-depth dialogue about their experiences with the goose harvesting program and their perspectives on how to utilize community strengths to enhance well-being. Including local knowledge is especially important when working with Indigenous and other marginalized communities due to the empowerment factors, especially accounting for the complex historic and present-day challenges these populations face. Overall, we acknowledge that health and well-being are more than just biomedical constructs and must be viewed through a social–cultural lens [47].

### 4.2. Food Security and Environmental Conservation

Goose harvesting has been a very important aspect of the *Omushkego* Cree culture in the James Bay region. Participants described the importance of goose harvesting as it provides a source of nutritious food and allows for knowledge transfer in the community. This knowledge was important as it allowed them to preserve teachings which were passed on from previous generations. One of the teachings discussed was the importance of sustainable harvesting to ensure that the waste from the animal harvesting was minimal. Within the community, another program utilized goose innards and bones, along with organic waste within the community, in a composter which aimed to minimize the dependency on imported fertilizers for crop production [95]. Although goose harvesting has many benefits, an overall decline in the goose harvesting was observed by participants, which was particularly of concern for future generations. For this reason, one of the major goals of this program was to revitalize the goose harvesting practices by connecting the Elders with youth and addressing barriers which have been identified in previous studies [3,10,13,92]. 

The present program also addressed issues of conservation. The importance of harvesting the overabundant goose species (i.e., lesser snow goose and giant Canada goose) for the conservation of ecosystems has been identified in many studies [10,17,23,30,96,97]. Studies have shown that high densities of lesser snow geese during snow-free seasons adversely impacts the local vegetation and land cover, causing desertification [96,97]. Thus, the suggestions for management of overabundant geese have included the increased harvesting of geese in the region to allow for habitat recovery [18,98]. Although the focus of the present program was on geese harvesting, there were concomitant benefits to the land from the harvesting of overabundant geese. Likewise, in Australia, the Healthy Country, Healthy People project aimed to integrate the reciprocal arrangement, “if you look after the country, the country will look after you” [99,100]. This concept is seen as critical for Indigenous well-being as it maintains cultural life, identity, and is associated with significantly better health [100]. 

Another important issue which we aimed to address through this study was food security. This was particularly important as Fort Albany FN has a very high prevalence of food insecurity [5]. Many participants expressed their concern for those experiencing food insecurity in the community, attributing it to a decline in the participation of traditional activities. There was an overall consensus amongst participants, even those who frequently consumed store-bought food, that traditional food was healthier. A study by Skinner, Hanning, Desjardins and Tsuji [4] found that the introduction of store-bought foods and the decreased acquisition of traditional foods has been detrimental for the nutritional health of Indigenous peoples. Indeed, a sharing-the-harvest intervention found that the consumption of snow goose led to higher intakes of protein, vitamin B12, iron, and zinc for youth [17]. It should be noted that all geese harvested in the intervention used non-toxic shotshells, as lead ammunition contaminates harvested game and lead ammunition has been identified as a major source of lead exposure for the James Bay Cree [4,92].

Participants in our study stressed the importance of revitalizing and maintaining traditional food practices for economic, cultural, and health reasons. Opportunities to maintain these practices were through the dissemination of Indigenous knowledge in the community, especially amongst youth. This knowledge was identified as being crucial for current and future generations, as it would provide them with the skills to obtain nutritious food from the land such as wild meats, berries, and other plants for medicine. Participants frequently discussed technology as impacting youth engagement, stating that they would prefer to be at home, as the community does not have cell service. Although technology was seen as a challenge, other participants described advances in technology making goose harvesting easier than in the past. One of these is the increasing use of freezers; many participants reported that this modernization allowed for the year-round access to traditional food, as goose and other meats could be stored and shared at a later date.

The sharing of traditional foods was mentioned by participants as an important way to combat food insecurity. In one study by Tsuji, Tsuji, Zuk, Davey and Liberda [3], the food sharing networks in the James Bay communities were documented; it was found that the network reached 76% of homes in one of the communities. In the present program, participants spoke of sharing amongst their families, Elders, and others in the community who were unable to hunt. By contrast, other studies have reported a decrease in food sharing in northern communities as a result of the high cost of hunting and an increasing number of households without hunters [4,101,102,103,104]. Participants in the present study discussed these factors and stated that the cost of obtaining supplies (i.e., boat, guns, ammunition) along with the lack of experienced hunters to teach youth and other community members, has resulted in a decrease in the harvesting. Many participants also discussed the cost of food and the overall rate of unemployment impacting their well-being and ability to access food. Indeed, research has identified one of the key determinants of food insecurity as being poverty, which is prevalent in many Indigenous communities which rely on social assistance [105]. Other programs aimed at subsidizing the costs of traditional activities, such as hunting and trapping, which have resulted in many benefits at the community level as well [106,107]. Thus, it is not surprising that many participants in our study reported their interests in maintaining community-led initiatives to share resources that would allow others to participate in the harvesting activities. This social cohesiveness highlights the strengths of the community and has been harnessed in other initiatives in the community. One type of initiative which has been successful in Fort Albany FN and other Indigenous communities is agricultural initiatives, mainly the use of community and individual gardens [10,87,108,109]. These community garden programs have been shown to increase food security and food sovereignty [87,109]. The support for this program within the community indicates that the policy initiatives which facilitate interactions with the land would be supported, as a means to promote the health and well-being of Indigenous populations [99]. The continuous implementation of policies and programs aimed at environmental conservation and health promotion, which are solely embedded in western science, undermine local expertise and knowledge, and further marginalize Indigenous populations [110]. Therefore, future collaborative efforts, including the recognition of Indigenous expertise, is crucial for efforts towards environmental conservation and improving the health and well-being outcomes.

### 4.3. Transferability

It has been suggested that the use of Indigenous harvesting programs in other parts of the world may be used to address overabundant and invasive species, and address food security [3,13]. Incorporating the harvesting of these species where food security issues could be addressed would not only allow for environmental conservation, but it would also provide cultural benefits to communities. One species which has been studied is the Asian Carp, which is a source of protein and omega-3 fatty acids, and therefore has been proposed to be used to address food security issues for low-income Americans [111,112]. Similarly, the invasive species known as target nutria (*Myocastor coypus*) in Louisiana, USA [113] and lionfish (*Pterois volitans*) [113,114] have been reported as examples of human consumption used to control invasive species. Studies have shown that the overall consumption of invasive species is a cost-effective solution, if coupled with appropriate policies, safeguards for consumption, and practices to ensure that the fish species is eradicated and not further cultivated for profit [113,114,115]. Similarly, in the Hawaiian Islands, wild boar species have a historical significance as a food source with traditions and rituals associated with their hunting; however, wild boars are well-known for their destructiveness, given their ability to modify entire ecosystems [116]. In summary, opportunities exist for harvest sharing programs of overabundant or invasive species to address food security issues among marginalized populations globally, including Indigenous peoples, with the benefits addressing issues beyond food security and environmental management.

### 4.4. Limitations

The limitations for this study included the limited sample sizes of the two programs; thus, the cortisol findings should be interpreted with caution. One reason for the limited sample size was that the pool of Elders and experienced people with the requisite knowledge was limited. As discussed previously, the pyramidal population structure of Fort Albany FN [54] in effect reduces the number of Elders or experts available for participation. As with the *Amisk* (beaver) harvesting program [13], the hunters and trappers were male. For the present program, participants were also required to possess a valid Possession and Acquisition License (PAL) to take part in the harvesting activities. Historically, females rarely possessed PALs; however, discussions with participants did acknowledge that this has changed, because of a licensing drive with the assistance of the Royal Canadian Mounted Police (RCMP) to help people of the western James Bay region obtain their PALs [3].

Another limitation of this study which was also identified by participants was that some only spoke “high” Cree. We attempted to overcome this barrier by inviting participants to bring a translator of their choice who spoke “high” Cree and English. However, some of the words and concepts were not directly translatable to the English language or the more commonly spoken conversational Cree. Any quotes which were translated were labeled “English Translation from Cree”. Lastly, due to the COVID-19 pandemic, several face-to-face meetings had to be canceled, with remote communication being used as the alternative.

## 5. Conclusions

The *Niska* (goose) harvesting program provided an opportunity for the Cree in Fort Albany FN to connect (or reconnect) with important aspects of their identity and increase social cohesion and well-being within the community. By harvesting overabundant geese, which are detrimental to the environment, not only was the issue of food security addressed, but it also contributed to the efforts to protect and provide time for the recovery of impacted ecosystems. Beyond these benefits, the harvesting program also addressed barriers which allowed for: the transfer of Indigenous knowledge; increased opportunities for healing; the strengthening of social networks (i.e., partnering youth with Elders); and other benefits associated with being on-the-land (i.e., feeling less stressed). Utilizing a local approach for on-the-land programs, especially in a remote community, allowed for the identification and adaptation to challenges using Indigenous knowledge.

Our findings highlight the need for the expansion of culturally relevant programs and policies for Indigenous peoples. Using approaches such as two-eyed seeing and CBPR allowed for more impactful and meaningful engagements with community members, which could be extended to other Indigenous and marginalized populations. Using a biomedical evaluation of cortisol has been successful in many non-Indigenous [62,117,118] and Indigenous studies [49,70,119]. However, the results from the present study highlight the importance of including Indigenous perspectives simultaneously. Through our discussions, many positive benefits were identified, which would not have been possible if solely the biomedical measure was utilized. Using the two-eyed seeing approach, which emphasizes the use of both the Indigenous and biomedical perspectives, is a robust approach for developing future on-the-land programs to address issues related to the well-being and environmental stewardship. As issues of food insecurity and overabundant/invasive species are becoming more apparent worldwide because of climate change [10,111], the present program may be adapted to other Indigenous or marginalized communities globally.

## Figures and Tables

**Figure 1 ijerph-20-03686-f001:**
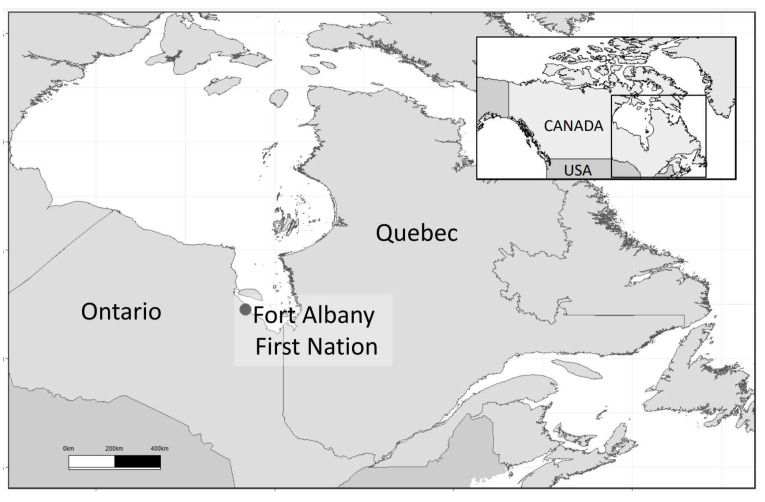
A map of Fort Albany FN, Ontario, CA.

**Figure 2 ijerph-20-03686-f002:**
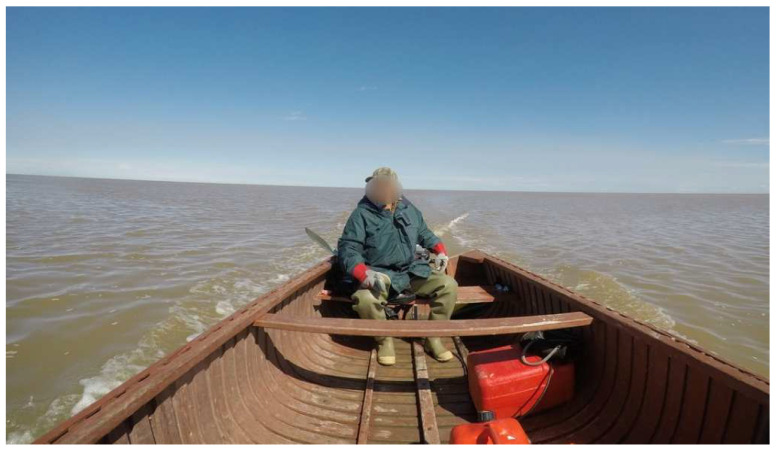
An Elder on a canoe guiding youth and other participants during the goose harvest.

**Figure 3 ijerph-20-03686-f003:**
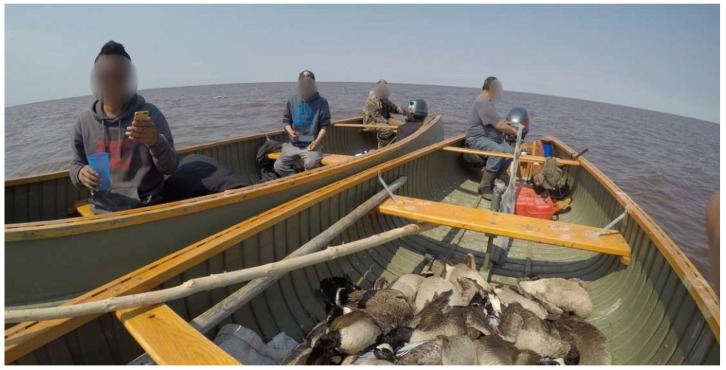
An Elder, on-the-land expert, and youth crossing paths in the bay while on canoes harvesting goose.

**Figure 4 ijerph-20-03686-f004:**
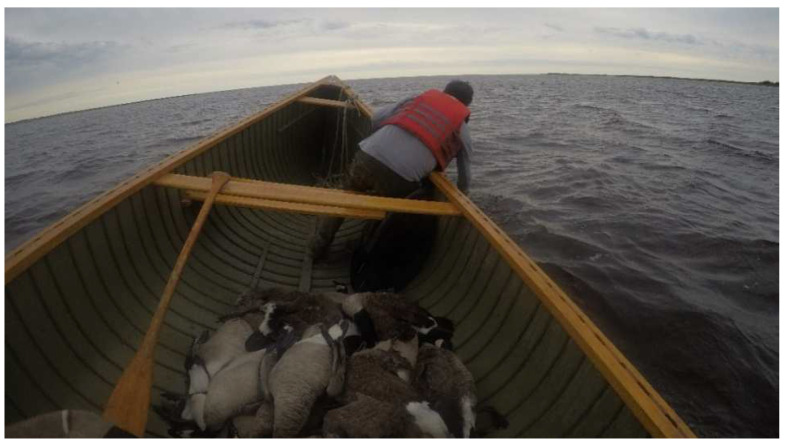
An on-the-land expert pulling a Canada goose from the bay into the canoe.

**Figure 5 ijerph-20-03686-f005:**
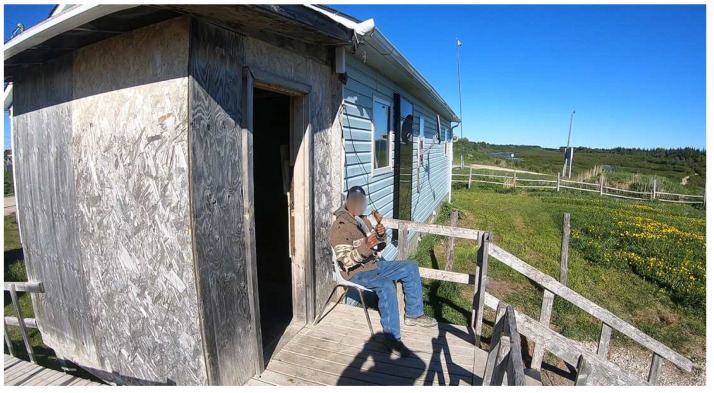
An on-the-land expert eating goose which was shared with him after the harvest.

**Table 1 ijerph-20-03686-t001:** Salivary cortisol (nmol/L) descriptives for pre- and post-participation in activities.

Project	Sample	Descriptives	
		*N*	Mean ± Standard Deviation (SD)
Goose Spring	Pre	13	6.52 ± 6.67
	Post	13	7.49 ± 6.23
Goose Summer	Pre	12	8.87 ± 7.56
	Post	12	6.51 ± 5.91

**Table 2 ijerph-20-03686-t002:** A summary of the thematic analysis.

Descriptive Themes	Subthemes	Analytical Themes
Knowledge		Learning from others; Sharing knowledge; Leading youth; Language
Identity		Being raised in the bush, Feelings of nostalgia
	Cultural Continuity	More programs needed; Culture to improve well-being; Technology barrier; Systemic barriers
Healing		Mental health; Keeping busy and active; Colonization; Substances; Healing; Traditional food; Traditional medicine; Trauma; Spirituality
Land		Emotions related to being on the land/community; Geese harvesting
	Food and Other Resources	Hunting for food and survival; Economic opportunities and challenges; Accessibility food/resources

## Data Availability

Not applicable.

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
