# Peer review of "Indigenous Land-Based Approaches to Well-Being: The *Niska* (Goose) Harvesting Program in Subarctic Ontario, Canada"

_ijerph, 2023, doi:10.3390/ijerph20043686_

Round 1
Reviewer 1 Report
The topic of the submitted manuscript is devoted to the very important issue of the contribution of the Indigenous knowledge and practice of goose harvesting to the community well-being including food security, ecosystem conservation, Indigenous identity and social cohesion of the Cree community in the Subarctic Ontario region. It brings a lot of interesting information regarding the perceptions of the members of the Cree regarding the above-mentioned issue. This case study is well written and brings clear message for related experts and decision makers.
However, there are some weaker points which could be improved.
In Introduction section, it is not articulated clearly, which “white space” in which theoretical concept will be filled in by the findings of this case study.
Brief description of the study area, with few sentences on geography of the James Bay region and the map with an indication of the Omushkego Cree community would be helpful in the Material and Methods section. The methodology description could be more complete. E.g. regarding informants (their age, gender, education, physical condition etc.) could be mentioned as well as the information how long the interviews approximately were. The authors should provide the complete list of leading questions used for interviewing informants. The quantitative and qualitative representativeness of the research sample (50) should be described and discussed more clearly. What is the basic set for the selection of this research sample, is that the whole population of the Omushkego Cree community (900)? As it is reflected in the Discussion section, it seems that the sample is too low to be quantitatively representative. It is also indicated that the qualitative structure of the sample does not reflect the population structure. So, the population structure could be briefly mentioned in the Material and Methods section.
In the Results section, the findings could be described in briefer and more structured way, mainly by comparing the results from the spring time with those from the summer time. The raw data collected in the frame of the qualitative research should be analysed (according to Ground Theory Method), not just simply presented (quoted). The comprehensive table containing meanings and their common categories discovered from qualitative research as well as design of scheme illustrating graphically the key findings of the qualitative research (e.g. the relations between categories) would be very helpful in the Results section.
The appropriateness of the salivary cortisol research could be touched in the Discussion section, while the significant problematic or beneficial cortisol changes are connected to the chronic stress which lasts for weeks to years while the duration of the harvest activities implemented in the frame of research was just from two days to week.
The Conclusion section should articulate summarization of the contribution of this research to the present-day theory. What is new in this study, which is not described in the cited or other relevant works?
Author Response
We thank the reviewer for their comments and have addressed their concerns and
yours, as detailed below. Note that additions and changes in text in the revised
manuscript appear as red text; deletions are not indicated. Please see the attachment.

Reviewer 2 Report
This paper uses community-based participatory research to conduct indigenous land-based approaches to well-being through the Niska (Goose) Harvesting Program in Subarctic Ontario, Canada. Authors contend that climate change has been detrimental to traditional harvesting practices among Indigenous communities already adversely impacted by colonization and colonial assimilative practices. Climate change has also disproportionately contributed to food insecurity among Indigenous peoples in Canada. Authors further assert the importance of integrating multiple perspectives to assess well-being among Indigenous peoples. The review is as follows:
1. The Introduction, including the first paragraph is very compelling and insightful.
2. Lines 55-57 – In “Some of the main factors contributing to this is the increased cost of food, increased dependence on market foods and the prohibitive costs of participating in land-based activities”, explain and provide a few examples of land-based activities.
3. Discussion within the Introduction of the significance of harvesting the Canada goose (Branta canadensis) is very informative and thought-provoking.
4. Within the study design, it is good to see the thoughtful consideration when using non-lead ammunition for this project.
5. Regarding study design, how was the study advertised and how were participants recruited? What was the response rate? (For example, did anyone decline participation?).
6. It is great to see the integration of the two-eyed seeing (Etuaptmumk) and community-based participatory (CBPR) approaches in this study.
7. The Discussion within the Results section of the declining use of the Cree language and interests of participants to preserve the language is insightful and poignant.
8. Consider bulleting the quotes in the paper to further illuminate them. Or the quotes can be placed within tables for organizational purposes.
9. Overall, this is a unique, insightful, comprehensive, and pertinent study. It is well-researched and would make a good contribution to the existing literature. It is pleasing to see a paper on this topic. The authors should be commended on their work. Minor suggestions are to expand discussion on the study recruitment and to arrange the quotes in a way that they can be further illuminated.
10. Check for appropriate self-citation as one of the authors seems to have an undue number of self-citations.
Author Response
We thank the reviewer for their comments and have addressed their concerns, as detailed below. Note that additions and changes in text in the revised
Manuscript appear as red text; deletions are not indicated. Please see the attachment.

Reviewer 3 Report
This is an interesting and promising manuscript but unacceptable in its current size - 36 pages.
Authors are encouraged to restructure and compress their text to an acceptable size. One can see the relevance of including lengthy, and often run on, quotes from interviews, but that should be done sparingly.
The main question revolved around the impact of climate change on goose farming amongst first-nation people in James Bay / Hudson Bay in Canada. The manuscript also provides extensive background and context information, i.e. colonial, neo-colonial and neoliberal developments and their adverse impact on indigenous communities.
The topic is extremely relevant, and manuscripts like this addressing the struggles and tribulations of indigenous communities is much needed. However, I am missing the rationale of this specific research and what/how it adds to the research domain. In other words, what is the paper's contribution, and to what field? Authors are encouraged to go beyond the " importance of using multiple perspectives when assessing well-being, especially in Indigenous peoples" and elaborate on what incorporating multiple perspectives means and how it could work.
I was baffled by the methodology. It combines quantitative and qualitative research methods, which is fine. However, it also feels that authors might have come to the same conclusions just by conducting interviews and observations. As a rule of thumb, mixed-method research provides a deeper understanding of a subject topic, as it relies on a large participant sample (p) and rich qualitative insights. In this case, P = 50, which in statistical terms might be seen as insignificant, is unclear how research participants were recruited and why.
The results and conclusions are self-referential. It seems that the authors had a result and conclusion in their mind and went on a quest to confirm their assumption. Which, in academic terms, is slightly problematic. Were there, for example, some unexpected findings that could enrich the analysis and discussion?
Another issue is the length of the results section. The authors use extensive quotes from the interviews, and given the nature of the research, that is fine. However, long quotes taking more than 3 lines is like using salt and pepper - it should be done sparingly or only when the authors find it inappropriate to paraphrase what respondents say.
References and styling are consistent. However, I did notice that prior publications of one of the authors are included opportunistically, making me question the present manuscript's originality.
I would advise authors to restructure and revisit their main argument and revise the manuscript thoroughly. It will also be beneficial if authors include a succinct and to-the-point reflections and limitations section of their research.
Author Response

(The authors gave the same response as above.)

Reviewer 4 Report
Thank you for the opportunity to review this interesting paper that discusses an important interface between Indigenous and Western scientific methods. It presents some novel and very interesting data and offers a well-written and important discussion of the challenge of listening to Indigenous testimony in the context of research that also focuses on measurement of specific biophysical indicators. My assessment is that there is some room for refinement, but the paper is strong and warrants publication with minor revision.
The paper pursues a two-pronged approach to assessing the effect on well-being of a program of goose harvest in the Omushkego Cree settlement of Fort Albany. I preface my commentary with an acknowledgement that I do not know the community or Cree cultural protocols and have approached my review task with a long background in interdisciplinary and geographical studies with First Nations across Australia.
The research integrated a two-eyed seeing (Etuaptmumk) and community-based participatory research approach, with a careful program to measure salivary cortisol, a biomedical measure of 20 stress, before and after participation in the program. The thrust of the paper is that participants’ self-evaluation of the value and well-being contribution of the program in terms of individual health and societal health was strong, but the measurement of cortisol did not exhibit statistically significant change. If one takes a First Nations centric position, this finding affirms the primacy of Indigenous knowledge as a continuing source of wisdom and guidance in everyday life and the incapacity of positivist science to evaluate the qualitative domain. Yet some of the writing continues to reflect the assumed primacy of the positivist framing of knowledge over the ontological constructions of relational belonging that underpins the CBPR reporting in the paper. For example, at lines 40-42 the paper says:
Traditional harvesting practices which were (and are) deeply rooted in familial and social systems of Indigenous communities were negatively impacted by colonization and colonial assimilative policies
The parenthetic comment normalizes the idea that harvesting practices are best thought as a relic of past traditions rather than normalizing them as a contemporary practice. I would suggest the authors consider rewording, and rethinking how this hidden privileging of positivist constructions of meaning are reflected elsewhere in the paper. Perhaps:
Traditional harvesting practices which are (and always have been) deeply rooted in familial and social systems of Indigenous communities continue to be negatively impacted by colonization and colonial assimilative policies
The lack of statistically significant reduction in cortisol measures in response to the program to affirm participants’ self-evaluated assessment that stress (and stresses) was reduced significantly by participation in the program is an interesting paradox. Of course the long-term and persistent nature of many of the issues of stress and trauma affecting participants (and the well-documented experience of PTSD and related concerns for many First Nations people as a result of ongoing trans-generational trauma such as the residential schools,
government policies and programs, structural racism, and the consequences of environmental and societal change) could perhaps be better discussed in the introductory parts of the paper to prepare readers for a more radical repositioning of the balance between the positivist and CBPR elements of the research. For example, I was reminded of the important Australian work of Judy Atkinson whose 2002 book Trauma Trails recognises that trauma has been embedded in cultural history for many First Nations people and needs to be addressed in healing the consequences of trauma. Similarly, the Australian experience has emphasised the important relationality of environmental and public health as the link between healthy country and healthy societies (eg Garnet et al 2009, Maclean etal 2013, Moorcroft et al 2012, Woodward et al 2019). Perhaps there is scope to reflect on this thinking and literature in framing the paper’s proposition in the early parts of the paper.
The paper’s great strength is the extensive interview quotations. These give readers a clear insight into the relational complexities facing Elders, ‘experts’ and ‘youth’ in negotiating changing relationality in Fort Albany. They reflect a context where social, political, economic, ecological and cultural changes constantly pull at familial, community and cultural relationships, obligations and protocols for all the participants (including the academic participants). These challenges are, of course, constructed across multiple scales from the micro-scale of negotiating access to food, fuel, medicine, support and time on a day-to-day basis through to negotiating adaptation to and mitigation of the consequences of planetary scale climate change. The participants’ eloquence and deep reflections reinforce the academic discussion, but could, I believe, be more strongly acknowledged as a source of adaptive relationality whose social and environmental ‘services’ should not be dependent on occasional academic projects for sustainability. Perhaps the authors could consider framing some policy recommendations around thinking about ‘payment for environmental services’ (see eg Bremer et al 2018, Muller 2008, Jackson & Palmer 2015). Similarly, the extensive emerging literature on Indigenous methodologies would suggest that some discussion of the governance of the research as integrating a two-eyed seeing (Etuaptmumk) and community-based participatory research approach is warranted. How did the academic participants comply with Omushkego Cree protocols – and how did the academic institutions respond to the need to work ‘with’ rather those protocols rather than treating the participants as research subjects (or objects!).
As I said above, thanks for this work. My comments will, I hope, encourage the team to take up and support the challenges of responding to the ontological pluralism of their research context and bring that challenge to their academic peers and funders. I applaud the work as making a valuable step in that direction.
Minor issues:
At line 971, I believe the word ‘participants’ needs to be added to the sentence.
I think some sub-headings need reformatting (eg bold not indented) to make the paper’s structure more clear.
References cited
Atkinson J (2002) Trauma trails, recreating song lines: the transgenerational effects of trauma in Indigenous Australia. North Melbourne: Spinifex Press.
Bremer LL, Brauman KA, Nelson S, et al. (2018) Relational values in evaluations of upstream social outcomes of watershed Payment for Ecosystem Services: a review. Current Opinion in Environmental Sustainability 35: 116-123.
Garnett ST, Sithole B, Whitehead PJ, et al. (2009) Healthy Country, Healthy People: Policy Implications of Links between Indigenous Human Health and Environmental Condition in Tropical Australia. Australian Journal of Public Administration 68(1): 53-66.
Jackson S and Palmer LR (2015) Reconceptualizing ecosystem services: Possibilities for cultivating and valuing the ethics and practices of care. Progress in Human Geography 39(2): 122-145.
Maclean K, Ross H, Cuthill M, et al. (2013) Healthy country, healthy people: An Australian Aboriginal organisation’s adaptive governance to enhance its social–ecological system. Geoforum 45(0): 94-105.
Moorcroft H, Ignjic E, Cowell S, et al. (2012) Conservation planning in a cross-cultural context: the Wunambal Gaambera Healthy Country Project in the Kimberley, Western Australia. Ecological Management & Restoration 13(1): 16-25.
Muller S (2008) Indigenous Payment for Environmental Service (PES) Opportunities in the Northern Territory: negotiating with customs. Australian Geographer 39(2): 149-170.
Woodward E and Marrfurra McTaggart P (2019) Co-developing Indigenous seasonal calendars to support ‘healthy Country, healthy people’ outcomes. Global Health Promotion 26(3_suppl): 26-34.
Author Response

(The authors gave the same response as above.)
